# Implicit In-Context Learning:
# Evidence from Artificial Language Experiments

**Xiaomeng Ma**[*]
Amazon
xiaomma@amazon.com

**Qihui Xu**
Department of Psychology
Ohio State University
xu.5430@osu.edu.

## Abstract

Humans acquire language through implicit learning, absorbing complex patterns without explicit awareness. While LLMs demonstrate impressive linguistic capabilities, it remains unclear whether they exhibit human-like pattern recognition during in-context learning at inferencing level. We adapted three classic artificial language learning experiments spanning morphology, morphosyntax, and syntax to systematically evaluate implicit learning at inferencing level in two state-of-the-art OpenAI models: gpt-4o and o3-mini. Our results reveal linguistic domain-specific alignment between models and human behaviors, o3-mini aligns better in morphology while both models align in syntax.

## 1 Introduction

Humans can acquire complex patterns implicitly through brief exposure, known as implicit learning, as evidenced by decades of artificial grammar and language-learning experiments in cognitive science (e.g. Reber, 1967; Saffran et al., 1996; Gómez & Gerken, 2000). Recent advancements in large language models (LLMs) have demonstrated similar remarkable linguistic capabilities, achieving human-like performance on diverse language tasks through in-context learning (ICL) - the ability to rapidly generalize from limited examples without explicit training updates (Brown et al., 2020; Webb et al., 2023). However, despite the striking parallels in outcomes, the cognitive and computational mechanisms underlying such rapid, example-based generalization in LLMs remain largely unknown.

Existing research on in-context learning in LLMs is fragmented and primarily application-oriented, often relying on heuristic methods such as prompt engineering or intuition-driven experimentation (e.g. Mavromatis et al., 2023; Ye et al., 2022). These approaches provide practical insights but fall short of elucidating underlying cognitive or computational processes. Concurrently, another line of research attempts to link transformer-based models to cognitive models, leveraging training or fine-tuning paradigms to explore representational similarities and cognitive plausibility (Goldstein et al., 2022; Binz & Schulz, 2023). However, these studies predominantly investigate learning mechanisms at the training or fine-tuning stages. Crucially, the process by which pre-trained models rapidly acquire structured linguistic knowledge during inference - when presented with limited examples - remains notably understudied.

Given this knowledge gap, the present study adopts an explicitly exploratory approach, systematically investigating whether LLMs exhibit implicit learning capacities analogous to human learners at the inferencing level. A key question of our paper is to ask: when task inputs are matched, do humans and models generate similar outputs in a similar way? Our approach is firmly grounded in well-established cognitive science paradigm. Specifically, we adapt three classical artificial language learning experiments that have provided robust insights into human implicit learning across different linguistic domains: morphology (Schuler et al., 2016; Schuler, 2017), morphosyntax (Valian & Coulson, 1988), and finite-state syntax (Alamia et al., 2020). By employing these paradigms, we aim to rigorously

---

[*]This work does not relate to the author's position at Amazon.

characterize foundational similarities and differences in pattern extraction mechanisms between human cognition and contemporary LLMs. We evaluated two state-of-the-art GPT models: gpt-4o (OpenAI et al., 2024), optimized broadly for general linguistic tasks, and o3-mini (OpenAI, 2025), fine-tuned specifically for explicit reasoning. Recent studies have indicated that reasoning-oriented models may underperform general-purpose models on tasks involving implicit statistical learning (Liu et al., 2024; Ku et al., 2025). While comparing reasoning-oriented and general-purpose models is not a primary goal of this study, observing their differences provides valuable insights into how implicit learning may vary across different model types. Our exploratory results indicate notable variability in model behaviors across the three implicit learning experiments without a clear overarching pattern. Methodologically, this study demonstrates the utility of cognitive science paradigms as rigorous tools for evaluating implicit learning in LLMs. Conceptually, our findings identify specific domains in which human and LLM cognitive processes align and diverge, offering groundwork for future hypothesis-driven investigations.

## 2 Related work

### 2.1 Implicit Learning: Insights from Human

Implicit learning research dates back to seminal work by Reber (1967), which demonstrated humans' ability to unconsciously extract patterns from artificial grammars. Subsequent research established that this capacity operates across various linguistic domains, from phonology to syntax, and is foundational to language acquisition (Saffran et al., 1996; Gomez & Gerken, 1999; Conway & Christiansen, 2006; Austin et al., 2022). Several factors influence implicit learning efficacy. Input frequency significantly impacts acquisition, with higher exposure facilitating stronger pattern recognition (Aslin & Newport, 2014; Valian & Coulson, 1988; Schuler et al., 2016; Schuler, 2017). Pattern complexity also matters; simpler regularities are more readily internalized than complex ones (Gómez & Gerken, 2000; Alamia et al., 2020).

Competing theoretical frameworks attempt to explain these processes, such as the statistical learning hypothesis (Saffran et al., 1996; Romberg & Saffran, 2010), the chunking hypothesis (Perruchet & Vinter, 1998) and the rule-based hypothesis emphasizes extraction of abstract algebraic patterns governing input structure (Marcus et al., 1999). While these perspectives offer valuable insights, debate continues regarding their relative contributions to implicit learning phenomena (Christiansen, 2019). Importantly, the present study takes an explicitly exploratory approach and does not presuppose any particular theory of implicit learning. Rather than engaging with theoretical debates about underlying mechanisms, we focus on empirically characterizing LLMs' behavior across various implicit learning tasks.

### 2.2 In-Context Learning in LLMs

In-Context Learning (ICL) (Radford et al., 2019) represents one of the most intriguing capabilities of LLMs, enbaling these models to adapt to new tasks without parameter updates. While the practical value of ICL is undeniable, its underlying mechanism remain frustratingly opaque (Mao et al., 2024). Studies often approach ICL from different aspects, examining isolated factors - pre-training data distribution (Chan et al., 2022), prompting strategies (Mavromatis et al., 2023; Ye et al., 2022), or model architectures (Dai et al., 2022; Cho et al., 2024).

The gap between ICL's practical success and theoretical understanding is particularly striking when compared to the methodical investigation of human implicit learning. Our study bridges this gap by adapting well-established experimental paradigms from cognitive science to systematically investigate implicit learning capabilities in LLMs. Rather than treating ICL as a monolithic ability, we examine specific learning processes across linguistic domains, providing a more principled approach to understanding how these models extract patterns from limited examples.

## 2.3 Cognitive science paradigms for probing LLM learning

As interest grows in comparing human and model cognition, researchers have increasingly turned to cognitive science methods to evaluate LLMs (Binz & Schulz, 2023). Classic paradigms in implicit learning—designed to isolate specific linguistic and cognitive processes—offer a controlled and interpretable framework for testing LLM behavior. Recent studies have adapted classic psychological methods to test the depth and flexibility of LLM learning, assessing whether models mirror human decision making, reasoning, and language processing (Binz & Schulz, 2023; Yasunaga et al., 2024; Goldstein et al., 2022). This approach has provided insights into both model capabilities and limitations. Our work contributes to this emerging methodological tradition by directly adapting three benchmark paradigms from the artificial language learning literature (Schuler et al., 2016; Valian & Coulson, 1988; Alamia et al., 2020)—which we introduce in detail below.

## 3 Experiment 1. Replicating Schuler et al. (2016)

Schuler et al. (2016) and Schuler (2017) investigated how adults and children implicitly generalize morphological patterns in an artificial language, specifically manipulating the ratio of regular to irregular plural noun types. [1] Their study contrasted categorical learning in children with probabilistic learning in adults by observing how learners applied a regular plural marker to novel nouns under different type-frequency conditions. We chose to replicate this experiment in large language models (LLMs) to evaluate whether models implicitly acquire morphological rules similarly to humans, and whether they generalize probabilistically or categorically. This replication thus provides valuable insight into the implicit morphological learning capabilities of LLMs and their similarity to human cognitive processes.

## 3.1 Experiment Setup

Schuler et al. (2016) created an artificial language experiment to examine morphological rule learning using 9 nonce (novel) words[2]. Words took either a regular plural marker (*-ka*) or irregular markers (e.g. *-po*, *-lee*, *-bae*). The study featured two conditions differing in regular-to-irregular proportions: 5R4E (5 regulars and 4 exceptions) and 3R6E (3 regulars and 6 exceptions). The nonce words roughly followed a Zipfian frequency distribution, with total 72 tokens (23 singulars and 49 plurals; see Appendix Table 2).

The human experiment used a picture-sentence matching paradigm, presenting participants with images of novel objects alongside sentences (e.g. singular 'Gentif mawg', plural: 'Gentif mawg*ka*', meaning 'there is/are [noun]'.) After learning 72 picture-sentence pairs, participants were tested on 6 new nonce words[3] to elicit appropriate plural markers in a sentence completion task (12 trials total with each test words appearing twice).

In our replication study, we adapted the experiment to a text-only format, preserving critical elements and frequencies from the original study. In the learning phase, we provided contextually rich paragraphs[4] containing both singular and plural nonce words, with 13 paragraphs per condition (see Appendix Table 4 for a complete set of paragraph templates). In the testing phase, LLMs completed fill-in-the-blank sentences, e.g. 'Where is my [noun]?' 'Which one are you talking about? You have [number] ___', producing plural markers for the same 6 testing words used in the original study.

The LLM experiments comprised 4 phases: 1) a brief introductory prompt, 2) a learning phase with paragraphs presented in random order, 3) a testing phase using the fill-in-

---

[1] The original study was testing Tolerance Principle (Yang, 2016), which is a theory that claims learning won't happen if the number of irregulars pass a threshold.

[2] These 9 words are: `mawg, tomber, glim, zup, spad, daygin, flairb, klidam, lepal`

[3] These 6 words are: `sep, norg, geed, daffin, fluggit, bleggin`

[4] For example, 'Joy only has one [noun]. Her sister has seven [noun][marker]. Her sister gave Joy four [noun][marker] so that each of them has four [noun][marker].'

the-blank prompts, and 4) a post-testing metalinguistic probe assessing explicit pattern awareness (see Appendix Table 3 for prompt template of each phase). To match the original study's 10 participants per condition, we conducted 15 runs per condition (3 random paragraph orders × 5 runs per order) on two GPT models: gpt-4o (temperature=0, top_p=1), and o3-mini model (reasoning effort=low)

## 3.2 Results

### 3.2.1 Regularization in the testing phase

Human adults showed similar regularization rates (percentage use of regular marker *-ka*) across conditions (5R4E: 65%, 3R6E: 51.7%, no significant difference). These rates closely match input token frequencies for *-ka* (5R4E: 75.5%; 3R63: 59.2%), suggesting probabilistic learning in adults. The LLMs exhibited divergent patterns. The o3-mini model mirrored adult behavior, showing robust regularization rates with no significant difference between conditions (5R4E: 75.6%; 3R63: 56.7%, p = 0.25), closely aligning with input token frequencies. In contrast, the gpt-4o model showed significantly lower and condition-sensitive regularization (5R4E: 41.7%, 3R6E: 5.0%; difference significant, p< 0.001), deviating significantly from human adults and input frequencies (both conditions p< 0.001).Table 5 in Appendix summarizes the regularization rates (percentage use of regular marker *-ka*) across conditions.

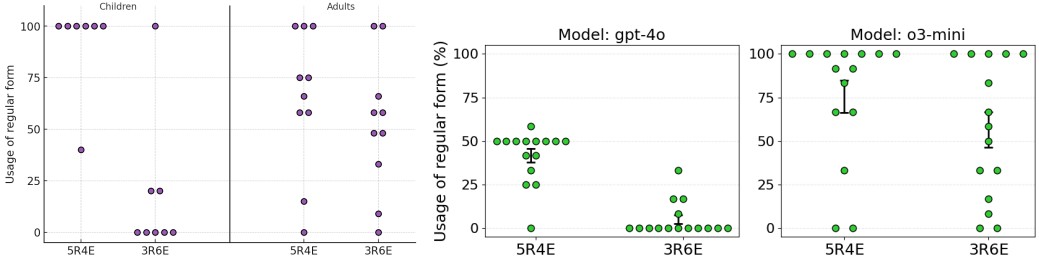

(a) Usage of the regular form by individual human participant. Reconstructed from Schuler (2017).

(b) Usage of the regular form by individual model runs, treating each run as an independent participant.

Figure 1: Comparison of individual participant's regularization across conditions.

Individual-level analyses (Figure 1) revealed adult participants varied considerably: some consistently applied the regular marker *-ka* in all testing trials, while others never or rarely used it. The o3-mini exhibited a similar bimodal distribution, suggesting probabilistic matching akin to adults. In contrast, the gpt-4o model showed a different pattern, with no runs achieving 100% regularization in either condition, and the 3R6E condition showed a drastic shift toward zero regularization, reflecting sensitivity to type-frequency proportions rather than probabilistic generalization.

These patterns can be further contextualized by considering children's data in the original study (as referenced in Table 5 and Figure 1). Children displayed categorical learning (significant condition difference, 5R3E: 91.7%, 3R6E: 16.9%), unlike adults' probabilistic pattern. The o3-mini model's behavior closely resembled adults' probabilistic strategy, whereas gpt-4o appeared to implement a distinct 'majority-type sensitivity', strong influenced by the proportion of regular to irregular word types.

### 3.2.2 Post-Testing Explicit Knowledge

We conducted a post-testing assessment of the models' explicit knowledge by asking 'How did you decide the blank word in the testing phase?' followed by 'Can you tell me more about the patterns you observed?'. We analyzed responses from all model runs and annotated the model's response on two dimensions: (1) whether they recognized the existence of any regular pattern (0=no, 1=yes) and (2) whether they correctly identified *-ka* as the

regular marker (0=no, 1=yes). Table 6 in Appendix summarizes number of model runs (out of 15 per condition) that demonstrated each type of knowledge. The post-testing explicit knowledge serves as supplementary evidence for interpreting the models' behavior. Given that models' self-explanations are known to have validity limitations (Madsen et al., 2024), we do not rely on them in isolation, but rather consider them alongside behavioral evidence.

The explicit recognition of the regular pattern varied substantially between models. The o3-mini model reliably identified the regular marker (*ka*) across conditions (10/15 in 5R4E and 7/15 in 3R6E). Explicit recognition strongly correlated with regularization performance in 3R6E condition (r=0.66, p<0.01)[5]. The gpt-4o model, in the contrast, struggled to recognize that a regular pattern existed (6/15 in 5R4E and 2/15 in 3R6E); and for the models that did, they never identified the regular marker correctly - some models misclassified both *-ka* and *-po* as regulars. Nevertheless, recognizing the existence of any regular pattern correlated with higher regularization rates in the 5R4E condition (r=0.53, p<0.05)[6].

These findings highlight a close alignment between implicit behavior and explicit awareness for the o3-mini model, in contrast to the limited explicit pattern understanding in the gpt-4o model, consistent with their differing implicit regularization behaviors.

## 4 Experiment 2. Replicating Valian & Coulson (1988)

Valian & Coulson (1988) examined how frequency influences the acquisition of context-dependent grammatical patterns through their artificial language study. Their experiment featured a miniature artificial grammar with two-phrase sequences (e.g., [aA bB]), where learners needed to master both word categories (distinguishing lower-case markers from upper-case content words) and their permissible combinatorial patterns. By manipulating the relative frequency of marker words across conditions, they examined how statistical properties affect the implicit learning of these morphosyntactic relationships. This experimental design serves as an ideal bridge between pure morphological learning (as in our first experiment) and pure syntactic acquisition (as in our third experiment), allowing us to investigate whether LLMs demonstrate human-like sensitivities to frequency when acquiring integrated morphosyntactic patterns.

### 4.1 Experiment Setup

Valian & Coulson (1988) developed an artificial language consisting of two adjacent phrases, each structured as a marker word followed by a content word, e.g. [aA bB] or [bB aA]. Marker words (a,b) always preceded their associated content words (A,B), creating dependency relationships. The experiment aimed to test whether participants implicitly learned these sequential ordering and specific marker-content associations under two frequency conditions. In the *high-frequency* condition, vocabulary consisted of two marker words and twelve content words, with markers appearing six times more frequently than content words. In the *low-frequency* condition, there were four marker words and six content words, with markers appearing 1.5 times more frequently. Each frequency condition also has 2 subconditions (S1, S2) varying which specific words served as markers or content words, as detailed in Table 3.

Participants were exposed to 24 training sentences presented in random order, followed by a testing phase comprising 96 sentences, divided into four trials (12 ungrammatical and 12 grammatical sentences for each trial). The ungrammatical sentences evenly included four error types per trial (3 sentences per type per trial): Type 1 - order violations (e.g. [aA Bb], [bB Aa]), Type 2 - category violations (e.g. [AA Bb], [BB aA]) Type 3 - single association violations (e.g. [aB bB] or [bB bA]) and Type 4: double association violations (e.g. [aB bA], [bA aB]). Participants judged sentence grammaticality without feedback.

---

[5]The correlation was not significant for 5R4E condition was probably due to the ceiling effect with high regularization rate across most runs.

[6]The correlation was not significant for 3R6E condition was probably due to the flooring effect, since most of the model runs had 0 regularization

In replicating this study with LLMs, we preserved the experimental structure closely: (1) an introductory prompt explaining the setup, (2) a learning phase with 24 grammatical sentences, (3) a testing phase where models judged 96 sentences divided into four trials, and (4) a post-testing phase explicitly probing the models' awareness of learned patterns (see Appendix Table 7 for the prompt templates in 4 phases). We tested two models: gpt-4o (temperature=0, top_p = 1) and o3-mini (reasoning effort = low), conducting 5 runs per model for each subcondition to match the original study's 5 human participants per subcondition.

## 4.2 Results

### 4.2.1 *Mean Errors Across Test Trials and conditions*

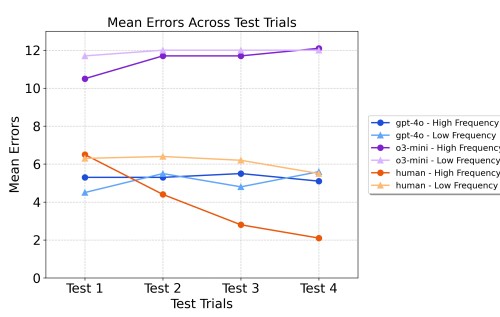

Figure 2: Mean error counts across test trials comparing human participants and models in high- and low-frequency conditions.

| Condition | Marker | Content words |
|---|---|---|
| High Freq. | a: alt
b: erd | A: deech, tasp, vabe, kicey, logoth, puser
B: hift, ghope, skige, cumo, fengle, wadim |
| | a: ong
b: ush | (same as above) |
| Low Freq. | a: alt, ong
b: erd, ush | A: puser, tasp, deech
B: ghope, hift, wadim |
| | a: alt, ong
b: erd, ush | A: vabe, kicey, logoth
B: skige, cumo, fengle |

Figure 3: Vocabulary across conditions/subconditions.

Figure 2 illustrates mean errors across testing trials for humans and models. Human participants exhibited clear learning effects: errors significantly decreased over trials, especially in the high-frequency condition. Humans also showed better overall performance in the high-frequency condition compared to low-frequency, confirming that frequency strongly influenced implicit learning.

In contrast, neither gpt-4o nor o3-mini demonstrated systematic learning improvements across trials, maintaining relatively constant error rates. Additionally, unlike humans, neither model showed sensitivity to frequency conditions. The gpt-4o model consistently performed better than the o3-mini model, achieving slightly fewer errors overall compared to human participants.

### 4.2.2 *Mean Errors by Error Type*

As summarized in Table 8, human participants made virtually no errors on simple sequence-related violations,(type 1: order violations and type 2: category violations), rapidly learning that there were two main categories of words (markers a, b and content words A, B) and the correct order of a sequence is [aA bB] or [bB aA]. However, they had difficulties with association-related errors (type 3 and 4), struggling to recognize the mapping relationship between a-A and b-B. The participants in high-frequency condition showed progressive improvement in the later trials, while in low-frequency condition continued to struggle. These results indicate human's ability to extract different linguistic patterns through exposure, with frequency playing a crucial role in learning complex patterns.

Models, however, showed a starkly different pattern. Neither model distinguished error types nor demonstrated improvements over time. Both models showed perfect precision (never incorrectly labeling sentences as grammatical) but low recall, consistently labeling grammatical sentences as ungrammatical. Specifically, o3-mini classified nearly all sentences as incorrect, while gpt-4o achieved moderate recall (~60%) for grammatical items. These

results indicate that models rely on rigid classification strategies rather than implicitly extracting grammatical patterns.

### 4.2.3    *Post-testing explicit knowledge anlaysis*

Explicit knowledge probing revealed further contrasts between humans and models. In the probing, we first asked the model about how they judged the grammaticality and explain the pattern they observed. Then we asked the model to correct ungrammatical sentences of 4 error types. All gpt-4o runs clearly identified markers versus content words, partially recognized word order (with some only accepted either [aA bB] or [bB aA] but not both), but consistently failed to grasp specific marker-content mappings. They succeeded only on the simpler sequence-related corrections (Types 1 and 2), mirroring their implicit classification behavior.

The o3-mini model adopted a fundamentally different approach, explicitly relying on memorizing training sentences rather than abstract rules. Many runs explicitly stated their inability to derive generalized rules, instead using an exact-match strategy, classifying any deviations from memorized examples as incorrect [7]. Notably, when asked to correct error sentences, o3-mini models succeeded in almost all tasks not by applying abstract patterns, but by referencing specific examples from the training data. This exemplar-based strategy directly explains their uniformly negative judgments during testing.

Together, these findings highlight fundamental cognitive differences between humans—who implicitly abstract grammatical patterns—and current LLMs, which either form incomplete abstractions or rely heavily on literal sentence memorization.

## 5    Experiment 3. Replicating Alamia et al. (2020)

Alamia et al. (2020) investigated pure syntactic learning through acquisition of finite-state grammars. Their study examined how participants learn abstract symbolic sequences governed by rule-based structures of varying complexity, demonstrating how statistical regularities at the syntactic level are implicitly extracted through exposure[8]. This experimental design completes our progression from morphological patterns (first experiment) through morphosyntactic relationships (second experiment) to pure syntactic structures. By replicating this paradigm with LLMs, we can evaluate whether these models demonstrate implicit statistical learning capabilities at the syntactic level—revealing potential similarities in how artificial and human intelligence process abstract grammatical patterns.

### 5.1    Experiment Setup

Alamia et al. (2020) tested participants' ability to learn two finite-state grammars of different complexity (Grammar A and Grammar B, as depicted in Appendix Figure 5 with Grammar A easier than Grammar B). Both grammars generate sequences of abstract symbols (e.g. 'X X V J' for Grammar A or 'M V R V V' for Grammar B). Each correct sentence had an incorrect counterpart created by changing a single letter (e.g. X X *T* J or M *X* R V V).

In the original study, participants learned grammar rules through 8 blocks of trials, each containing 30 pairs of correct and incorrect sentences (480 total sentences). They were informed that half of the sentences are incorrect. In the learning task, the participants provided responses indicating sentence correctness, and received immediate feedback. Following learning, implicit grammatical knowledge was assessed using a 7-question post-test questionnaire (shown in Appendix Table 10).

---

[7] For example, when asked about how they judged the grammatically of the test sentences, some model answered: '...Without any guiding rules, the simplest model was that a "correct" sentence must match one of the examples (and any slight reordering counts as a deviation).'. When asked about what patterns they observed some model answered: 'I did not derive a concise or precise set of rules... My responses were based on an overall impression rather than an explicit, rule-based system.'

[8] In their original study, they also trained feedforward and recurrent neural network from scratch to compare the models' behavior with human behavior.

To experiment with LLMs, we reduced each trial to 6 blocks, each with 10 pairs of correct and incorrect sentences (total 120 sentences). Our LLM experimental procedure involves 4 phases: (1) a introductory prompt, (2) a learning phase where the model judged each sentence and received dynamic feedback, (3) a post-testing phase of implicit knowledge assessment through questionnaires. The prompt template for each phase is shown in Appendix Table 9. We tested two models: gpt-4o (temperature = 0, top_p = 1) and o3-mini (reasoning effort = low), conducting 15 runs (3 random seeds to generate sentences for each grammar × 5 runs) per model for each grammar to match the 15 human participants in the original experiments.

## 5.2 Results

### 5.2.1 Accuracy Over Learning Blocks

The accuracy across learning blocks for human and models is shown in Figure 4. The Bayesian repeated measures ANOVA revealed extremely strong evidence for learning effects across blocks for both models (BF ≫ 100), closely mirroring the human performance reported in Alamia et al. (2020) (BF ≫ 100). We also found strong evidence that Grammar A was easier to learn than Grammar B for both models, replicating the findings in human participants and confirming that grammatical complexity impacts learning similarly across both humans and LLMs. Additionally, gpt-4o generally achieved higher accuracy than o3-mini across both grammar types (BF = 4.63).

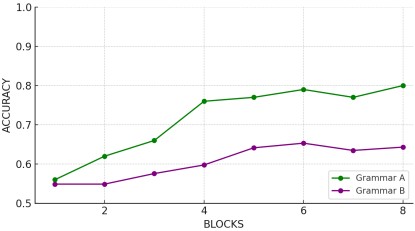
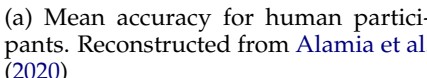
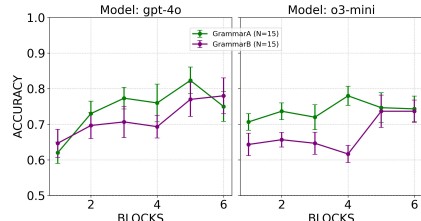

(a) Mean accuracy for human partici-pants. Reconstructed from Alamia et al. (2020)

(b) Mean accuracy for language models across 6 blocks

Figure 4: Comparison of mean accuracy between human participants and models.

### 5.2.2 Post-testing Implicit Knowledge Questionnaire

Following the learning blocks, we tested the models with the same 7 questions used with human participants to assess their implicit knowledge of the grammar rules. While human participants responded through multiple-choice questions, we used open-ended format for LLMs and evaluated their responses for correctness. The results showed strong evidence that both humans and LLMs developed better implicit knowedge for the easier Grammar A than Grammar B. On the questionnaires, human participants demonstrated significantly higher sensitivity for Grammar A (BF > 100), while LLMs showed substantially higher accuracy for Grammar A (BF >260) (illustrated in Appendix Figure 4). Although the gpt-4o's accuracy was numerically higher than o3-mini's, this difference did not reach statistical significance. These findings suggest that LLMs and humans show similar patterns of implicit statistical learning, with both performing better on the simpler Grammar A and demonstrating progressive improvement over training trials.

## 6 Discussions

### 6.1 Parallels and Divergences with Human Learning

Our experiments provide novel insights into the implicit linguistic capabilities of large language models. Across three distinct linguistic domains, we observe intriguing patterns

| | | Similar to Human Patterns? | |
| Domain | Conditions | gpt-4o | o3-mini |
|---|---|---|---|
| Morphology | 5R4E | ✗ | ✓ |
| (word level) | 3R6E | ✗ | ✓ |
| Morpho-syntax | high-freq | ✗ | ✗ |
| (phrasal level) | low-freq | ✓ | ✗ |
| Syntax | easy | ✓ | ✓ |
| (full sequence) | complex | ✓ | ✓ |

Table 1: Summary of models performance (similar to humans or not) in different experiments.

of similarity and difference between human learning and LLM behavior (summarized in Table 1.

Most notably, our findings reveal domain-specific alignment between models and human cognition. In Experiment 1, morphological implicit learning revealed probabilistic generalization in o3-mini similar to human adults, whereas gpt-4o demonstrated pronounced type-frequency sensitivity. This suggests that the o3-mini model's reasoning-enhanced design may enable more implicit statistical learning for morphological tasks, since deriving a regular pattern in this particular task doesn't fully rely on statistical majority.

In Experiment 2, the gpt-4o model behaved similar to humans in the low-frequency condition; however, did not exhibit better performance in high-frequency condition, suggesting that the model is not sensitive to frequency manipulation at morpho-syntax level. In contrast, the o3-mini model showed rigid or exemplar-based strategies rather than dynamically implicit pattern extraction. This divergence suggests a fundamental difference in how context-dependent grammatical patterns are processed. Unlike humans, who gradually abstract increasingly complex dependencies through exposure, both models appeared to rely on more brittle strategies.

In Experiment 3, both models closely mirrored human learning trajectories in the finite-state grammar learning task. This alignment in syntactic pattern learning, particularly the shared sensitivity to grammatical complexity, suggests that sequence learning mechanisms may be more fundamentally shared between human cognition and ICL in LLMs. The ability to recognize and generalize abstract sequential patterns without semantic content appears to be equally attainable through different cognitive architectures. Our findings point to a more nuanced reality where alignment with human learning depends on specific linguistic domains and learning mechanisms. This perspective helps explain contradictory findings in previous literature, where studies focusing on different linguistic phenomena have reached divergent conclusions about LLMs' similarity to human learning.

## 6.2 Implications and Future Directions

Our findings revealed linguistic domain-specific patterns of alignment between human and LLM learning, suggesting that these LLM models could serve as valuable computational implementations of certain cognitive theories, particularly in domains like sequence learning and morphological generalizations. These divergences, especially when task inputs are matched, underscore that LLMs and humans may not be solving the same computational problems or using similar reasoning procedures—highlighting the importance of distinguishing between surface-level behavioral alignment and deeper representational similarity. In addition, the striking differences between o3-mini and gpt-4o across experiments suggest that optimization for explicit reasoning may sometimes come at the cost of implicit pattern recognition capabilities that more closely resemble human statistical learning.

Future research should systematically investigate the architectural features that enable more human-like implicit learning across linguistic domains. Specifically, three directions appear promising: (1) developing benchmark datasets based on cognitive science paradigms

to standardize evaluation of implicit learning capabilities across models; (2) exploring how pre-training data characteristics affect the emergence of different learning strategies; and (3) implementing controlled variations in model architecture to isolate components that contribute to specific implicit learning abilities. Additionally, examining functional language use alongside formal competence will be key to fully understanding implicit learning parallels between humans and LLMs.

This study remains exploratory, deliberately avoiding commitment to specific theoretical hypotheses at this stage. Future targeted studies should also build on our exploratory findings, formulating and testing explicit hypotheses concerning specific linguistic phenomena. This study focuses on two OpenAI models—`o3-mini` and `gpt-4o`—which limits generalizability. Future work should examine whether the observed patterns hold across models from different developers and scales to assess whether these behaviors reflect general properties of language models or model-specific traits. Ultimately, by bridging cognitive science methodologies with computational linguistics, we contribute a crucial foundation for understanding how implicit learning unfolds both in humans and artificial language models.

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

## Appendix

| Rank | Frequency | N plurals | Noun | 5R4E marker | 3R6E marker |
|------|-----------|-----------|------|-------------|-------------|
| 1 | 24 | 16 | mawg | ka | ka |
| 2 | 12 | 8 | tomber | ka | ka |
| 3 | 8 | 5 | glim | ka | ka |
| 4 | 6 | 4 | zup | ka | po |
| 5 | 6 | 4 | spad | ka | lee |
| 6 | 4 | 3 | daygin | po | bae |
| 7 | 4 | 3 | flairb | lee | tay |
| 8 | 4 | 3 | klidam | bae | muy |
| 9 | 4 | 3 | lepal | tay | woo |

Table 2: The frequency distribution of each nonce word in the Schuler et al. (2016)

| Phase | | Prompt Template for replication Schuler et al. (2016) |
|---|---|---|
| Starting | | Let's play a game. In this game, you'll see words from an artificial language. These words have no meaning. The game consists of a learning round and a testing round. In the learning round, I will present you some sentences containing the words from this artificial language. In the testing round, I will present you some more sentences containing the words from this artificial language and you'll need to fill in the blank. Last, I will ask you some questions about this game. |
| Learning | | Now it's the learning round. I will present you some sentences in this artificial language. You can acknowledge by replying 'I'm ready for the testing round.' {paragraphs} |
| Testing | | Now it's testing round. Fill in the blank for the following sentence, just reply the word for the blank: 'Where is my {noun}?' 'Which one are you talking about? You have {number} ___.' |
| Post-testing | 1 | OK. Now the testing round is over. Let's reflect on this game. How did you decide the word for the blank in the testing round? |
| | 2 | Can you tell me more about the patterns you observed? |

Table 3: Prompt Templates for replicating Schuler et al. (2016)

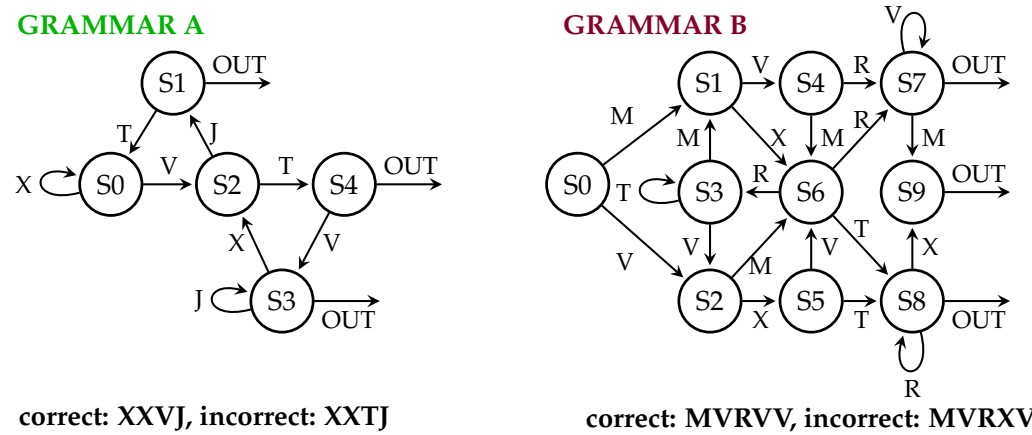

**GRAMMAR A**

correct: XXVJ, incorrect: XXTJ

**GRAMMAR B**

correct: MVRVV, incorrect: MVRXV

Figure 5: Finite-state for Grammar A and Grammar B in Alamia et al. (2020)

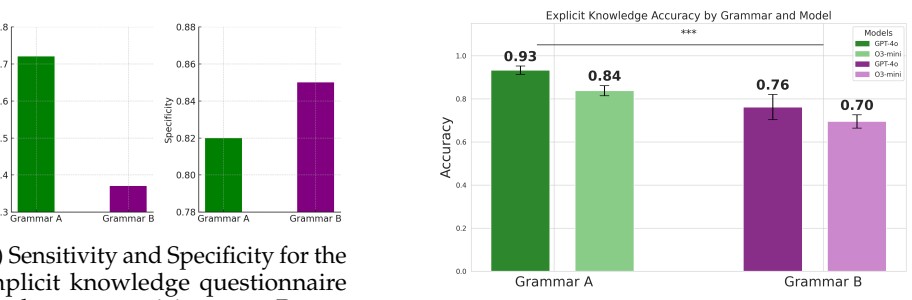

(a) Sensitivity and Specificity for the implicit knowledge questionnaire for human participants. Reconstructed from Alamia et al. (2020)

(b) Accuracy for the implicit knowledge questionnaire for models

Figure 6: Comparison of results for implicit knowledge questionnaire. Both humans and models showed better implicit knowledge with Grammar A than Grammar B.

| Noun | Paragraph Template |
|---|---|
| mawg | Peter bought six mawg[marker] from the store. He got home and opened the pack only to find five mawg[marker]. He found his backpack again making sure he didn't miss one mawg. So he went back to the store to get the missing mawg. The shopkeeper was really sorry that he forgot one mawg and ended up giving Peter two more mawg[marker]. Now Peter has seven mawg[marker]. |
| mawg | Mia found seven mawg[marker] in the basement. Only one mawg was intact and the other six mawg[marker] were broken. Mia decided to use the broken parts of the mawg[marker] to make a new mawg. In the end, she succeefully restored two mawg[marker] with the six broken mawg[marker]. |
| mawg | Alex has nine mawg[marker] and Lily has six mawg[marker]. They are trying to see if they can exchange mawg[marker] to make each other has the same number of mawg[marker]. However, if Alex gives one mawg to Lily, he still has more mawg[marker] than Lily. If Alex gives two mawga[marker] to Lily, then Lily will have one more mawg than Alex. In the end, Alex decides to give one mawg to Lily and Lily decides to buy one mawg. So each of them will have eight mawg[marker]. |
| tomber | Yulia counts six tomber[marker] on her shelf. There are two red tomber[marker], two green tomber[marker], one yellow tomber and one pink tomber. She also wants two orange tomber[marker] to finish her collection. |
| tomber | Tom has four tomber[marker] and Jill has seven tomber[marker]. If Jill gives Tom one tomber, Jill still has more tomber[marker] than Tom. If Jill gives Tom two tomber[marker], Tom would have one more tomber than Jill. |
| glim | Frank bought two red glim[marker] for Penny, but she actually wanted green glim[marker]. Luckily, Nina brought a green glim[marker] for her. |
| glim | Bob has three glim[marker]. He gave two glim[marker] to Ben and one glim to Nola. Now he only has one glim. Bob wants to buy one more glim. |
| zup | Katie has eight zup[marker]. Kerry only had one zup. Kerry asked if Katie can give her three zup[marker]. Katie said no, but she can give her two zup[marker]. Kerry agreed, thinking that three zup[marker] is better than one zup. |
| spad | John bought four spad[marker] for his art project. He thought he might need more spad[marker] but he only used one spad. He tried to return the three unused spad[marker] but only successfully returned one spad. He didn't know what to do with the other two spad[marker]. |
| daygin | Joy only has one daygin. Her sister has seven daygin[marker]. Her sister gave Joy four daygin[marker] so that each of them have four daygin[marker]. |
| flairb | There are six flairb[marker] on the table. John took two flairb[marker]. Helen took three flairb[marker]. Mike took the only one flairb left. |
| klidam | Sally had five klidam[marker]. She gave Anne two klidam[marker] and gave Susanne two klidam[marker]. Now she only have one klidam. |
| lepal | Mark used to have four lepal[marker]. He gave one lepal to his sister. Mark now have three lepal[marker]. His brother also asked Mark to give him two lepal[marker] but he said no. |

Table 4: Paragraph Template for replicating Schuler et al. (2016)

| | 5R4E (%) | 3R6E (%) | Difference |
|---|---|---|---|
| Human adults | 65.0 | 51.7 | 13.3 |
| gpt-4o | $41.7 \pm 8.0$*** | $5.0 \pm 3.0$*** | 36.7*** |
| o3-mini | $75.6 \pm 6.5$ | $56.7 \pm 7.0$ | 18.9 |
| Human children | 91.7 | 16.9 | 74.8*** |

Table 5: Regularization rates across all test trials for human adults, children, and LLMs. For the 5R4E and 3R6E columns, significance markers indicate statistically significant differences from human adult performance. For the Difference column, markers indicate significant differences between conditions within each model/group. Children's data (bottom row) is presented for reference only and was not used for statistical comparisons with the models. Humans data are from Schuler (2017).

| Model | Condition | Recognized Regular Pattern | Correctly Identified -ka as Regular |
|-------|-----------|---------------------------|-------------------------------------|
| gpt-4o | 5R4E | 6* | 0 |
|        | 3R6E | 2 | 0 |
| o3-mini | 5R4E | 10 | 10 |
|         | 3R6E | 9 | 7** |

Table 6: For Experiment 1 replicating Schuler et al. (2016), models' explicit knowledge of morphological patterns across conditions. Numbers indicate count of model runs (out of 15) demonstrating each type of knowledge. *indicates significant corelation with regularization rate.

| Phase | | Prompt |
|-------|---|--------|
| Starting | | Let's play a game. In this game, you'll see sentences from an artificial language, that composed of nonsense words that had no meanings. The game consists of the one learning round and four testing rounds. In learning round, I will present you some sentences in this language. You can simply acknowledge by saying 'I'm ready." In the testing round, I will present you some more sentences and you'll need to judge if these sentences are correct or incorrect. In the testing round, all the sentences are presented one by one and you need to answer one by one. Last, I will ask you some questions about this game. |
| Learning | | Now it's the learning round. I will present you some sentences in this artificial language. You can acknowledge by replying 'I'm ready for the testing round'. {training_sentences}. |
| Testing | start | OK. Now it's testing round {number}. Judge whether the test sentence is correct or incorrect in this artificial language. Answer 'correct or incorrect'.{test_sentence}. |
|  | end | OK. Now test round {number} is over. Take some time to reflect. We are about to start the next round. |
| Post-testing | 1 | OK. Now all test rounds are over. Let's reflect on this game. How did you judge whether a sentence is correct or incorrect? |
|  | 2 | Can you tell me more about the patterns you observed? Let's review some answers. {incorrect_sentence} is incorrect. Can you tell me why it's incorrect? How would you fix it to be correct? |

Table 7: Prompt Templates for Replicating Valian & Coulson (1988)

| Error type[a] | Test | | | | |
|---|---|---|---|---|---|
| | I | II | III | IV | Total |
| High frequency | | | | | |
| 1 | .1 | 0 | 0 | 0 | .1 |
| 2 | .1 | 0 | 0 | 0 | .1 |
| 3 | 2.8 | 2.1 | 1.3 | 1.0 | 7.2 |
| 4 | 2.5 | 1.4 | .7 | .6 | 5.2 |
| All false | | | | | |
| positive (1–4) | 5.5 | 3.5 | 2.0 | 1.6 | 12.6 |
| False negative | 1.0 | .9 | .8 | .5 | 3.2 |
| Total errors | 6.5 | 4.4 | 2.8 | 2.1 | 15.8 |
| Low frequency | | | | | |
| 1 | 0 | 0 | 0 | 0 | 0 |
| 2 | .1 | 0 | 0 | 0 | .1 |
| 3 | 2.3 | 2.8 | 2.6 | 2.0 | 9.7 |
| 4 | 2.4 | 2.7 | 2.3 | 1.7 | 9.1 |
| All false | | | | | |
| positive (1–4) | 4.8 | 5.5 | 4.9 | 3.7 | 18.9 |
| False negative | 1.5 | .9 | 1.3 | 1.7 | 5.4 |
| Total errors | 6.3 | 6.4 | 6.2 | 5.4 | 24.3 |

Note. For each error type, maximum score = 3; for false negatives, maximum = 12.
[a] For description of error types, see text.

Table 8: Human's mean error counts by error types. Data from Valian & Coulson (1988).

| Phase | | Prompt Templates for Replication Alamia et al. (2020) |
|---|---|---|
| Starting | | Let's play a game. In this game you will see sentences from an artificial language. There are only {vocabulary_size} words in this language: {vocabulary}. They don't have meanings. I will show you several batches of sentences. Each batch contains 20 sentences – half of the sentences are grammatical in this artificial language and half of the sentences are not. You need to guess if the sentence is grammatical or not. I'll provide you feedback for each sentences so that you can use the feedback to improve your guess. |
| Learning | start | Guess the following sentence is grammatical or not. Only output yes or no. {current_sentence}. |
| | middle | You answer is {correct_or_incorrect}. {previous_sentence} is {grammatical_or_ungrammatical}. Guess the following sentence is grammatical or not. Only output yes or no. {current_sentence}. |
| | end | You answer is {correct_or_incorrect}. {previous_sentence} is {grammatical_or_ungrammatical}. Now it's the end of {batch_n}. We are about to start {batch_n+1}. |
| Post-testing | | Now it's the end of learning phase. Let's take some time to reflect on this game. I will ask you 7 questions and you need to simply provide the answer to these questions. {questions} |

Table 9: Prompt Templates for replicating Alamia et al. (2020)

1. Which letter(s) is(are) more likely to be in the first position?
2. Which letter(s) is(are) more likely to be in the last position?
3. Which letter(s) is(are) more likely to be in the second position?
4. Which letter(s) cannot be presented twice consequently (e.g. in position 2 and 3, or in position 3 and 4, etc)?
5. Which letter(s) is(are) more likely to appear after the letter 'X'?
6. Which letter(s) is(are) more likely to appear after the bigram 'XT'?
(For Grammar A) 7. Which letter(s) is(are) more likely to appear after the bigram 'XV'?
(For Grammar B) 7. Which letter(s) is(are) more likely to appear after the bigram 'MX'?

Table 10: Questionnaires for Grammar A and Grammar B in Replicating Alamia et al. (2020)

