# OpenReview forum: "Implicit In-Context Learning: Evidence from Artificial Language Experiments"
_colmweb.org/COLM/2025/Conference — COLM 2025_

### Official Review · Reviewer_CGzC · 2025-04-27

**Rating:** 6
**Confidence:** 4
**Ethics Flag:** 1

**Summary:**

This paper systematically investigates whether LLMs exhibit human-like implicit learning capabilities during in-context learning by adapting three classic artificial language learning experiments spanning morphology, morphosyntax, and syntax. The work is well written with clear motivations and experiments. It adapts existing studies on humans to LLMs and reveals domain-specific similarities between LLMs and humans, which could provide insights into both human cognition and artificial intelligence mechanisms.

**Questions To Authors:**

In general, I really enjoyed reading this paper :) and would appreciate it if the authors could provide further justifications for the points I commented on.

**Reasons To Accept:**

1.  This is a very interesting topic. More and more scientists (especially in cognitive science and neurolinguistics) have been applying LLMs to test human theories. Therefore, it is significant to understand whether LLMs behave similarly to humans.

2. The authors take a novel perspective on ICL and provide new insights compared to previous ICL work.

3. The experiments are clearly written and well presented.

**Reasons To Reject:**

1. I am concerned about drawing parallels between human implicit learning and LLM in-context learning, which are mechanistically distinct processes. Human implicit learning involves gradual, unconscious pattern extraction over time with memory consolidation, while LLM ICL occurs in a single forward pass without temporal dynamics or lasting memory traces.

2. Given the fundamental differences noted above, the observed similarities or divergences may not truly reflect meaningful cognitive parallels or offer substantive insights into either human cognition or LLM mechanisms.

3. Only OpenAI models are tested. Including models from other developers and across different scales (especially smaller models) would help establish whether the observed patterns represent general properties of language models and whether it is consistent across different scales or only specific to some large models, which may emerge stronger ICL abilities.

4. Testing only two models with distinct patterns is insufficient for drawing general conclusions or making statistically significant statements about model behavior

4. The paper focuses exclusively on formal linguistic competence, an area where LLMs traditionally excel. A comprehensive understanding of implicit learning parallels would require examining functional linguistic competence as well.

---

> ### Author Response · Authors · 2025-06-03
>
> Thanks so much for your thoughtful review and constructive feedback. We'd like to address each concern as follows:
>
> Point 1 and 2. Mechanistic differences between human implicit learning and LLM ICL
>
> We acknowledge and fully agree that human implicit learning and LLM in-context learning: operate through fundamentally different mechanisms, particularly regarding memory consolidation and temporal dynamics as you pointed out.  However, our study wanted to deliberately focus on the statistical processes underlying both systems. The three studies we replicated all involves manipulation of input frequency, and we wanted to focus on the functional statistical learning patterns, instead of making any claims about the deep mechanisms of learning. By examining whether LLM and humans exhibit similar sensitivity to statistical regularities, we can identify where the LLM and the human differs. This can guide us to where to investigate further about the different statistical processes. For example, in experience 1 the o1 model showed similar token frequency sensitivity as human adults, while the 4o model showed a different profile. This functional statistical learning comparison serves as a crucial step for developing more sophisticated probes of the LLM capacities based on human learning research.
>
> Point 3 and 4. Limited model coverage
>
> We appreciate this important limitation! Our choice of OpenAI models was indeed resource-constrained. Moreover, our study was explicitly designed and presented as an exploratory investigation to establish a methodological foundation for comparing human implicit learning with LLM ICL. Even with two models, we already revealed several different patterns between models and human.  For our next steps, we definitely will test different scaled models, especially smaller models, to reveal more interesting results. In addition, we would like to dive deep with a specific aspect in implicit learning with more models, so that we can draw more definitive conclusions.
>
> Point 5. This is an excellent point! Thanks very much for bringing up! In this paper, we only replicated studies that address formal linguistic competence, as these tasks are well-established in human experiments. Investigating the implicit learning in functional linguistic competence and comparing to ICL is definitely an interesting and exciting next step. We are planning to review the literature on implicit learning in functional contexts and designing experiments that can meaningfully test both human participants and LLMs on pragmatic and communicative aspects of language learning.
>
> Thanks again for your review!

---

> > ### Comment · Reviewer_CGzC · 2025-06-06
> >
> > Thank you for your response. I still find the current results somewhat limited in terms of generality and effectiveness. Therefore, I will maintain my current (positive) score. I would consider increasing the score if the authors provide additional results on a broader range of models (smaller open-source models such as LLaMA and Qwen shall be fine) or offer further insights into Point 5.

---

### Official Review · Reviewer_FRjt · 2025-05-11

**Rating:** 7
**Confidence:** 5
**Ethics Flag:** 2

**Summary:**

This paper compares the behavior on GPT-4o and o3-mini on modified versions of three artificial language learning tasks. It finds that the models behave in many ways differently from each other and from humans in the original experiments, though there are some similarities

**Ethics Concerns Details:**

Only a minor thing. The paper uses probably copyrighted images from other papers in Figure 1 and 3. They should produce new figures with the original numbers as they did in Figure 2.

**Questions To Authors:**

Some suggestions -

Furlong et al. 2022 is a good summary paper that could be added to Sec 2.1

Schuler 2017 argues for a rule generalization framework and could be cited in the second paragraph of 2.1

If you would like this paper to reach a cogsci audience, you need to add a paragraph explaining what is meant by in-context learning in 2.2.

**Reasons To Accept:**

The paper is to-the-point and easy to follow at a high level (except also see below). It is a straightforward direct application of methods from cognitive science to the study of LLMs. It's a nice straightforward contribution to the field in the style of an ACL short paper. I like that the paper engaged with the cog sci literature and looked at specific ALL studies.

**Reasons To Reject:**

While I basically like the paper, it has a couple of faults.

First, it's exposition felt unclear and messy in certain spots. In 3.1, the experimental setup was unclear. You use a modified version of the paradigm where you replace the image presentation with text. Fine. Is this text a direct "translation" of the images, or are the carrier phrases altogether different? This choice could substantially affect the results, because, just as with humans, a different data presentation during the trial might yield very different results. It's a problem of reproducibility.

In the second experiment, you discuss a few different types of error stimuli. How were the errors presented in the experiment? One type per trial? All interleaved? How did the models perform regarding each error type?

Second, the paper makes a convincing case for using methods from cogsci to help us understand LLMs, but in the conclusion of the paper, they propose doing the opposite - using LLMs as tools for cognitive science "LLM models could serve as valuable computa-
356 tional implementations of certain cognitive theories" I don't understand the basis of this argument, since the models clearly behaved differently from humans. If anything, we shouldn't use them in cogsci for these tasks.


In Figure 1 and 3, you show copies of the plots that appeared in original papers. Those images are typically copyrighted, and the psych/cogsci communities take that more seriously than NLP/ML. I don't think you can use those plots without permission from the authors. You will need to create new figures using the original numbers as you did in Figure 2. This is technically an ethics issue, albeit a minor one that is easy to correct, so I have to flag it. I see this issue as an example of the "messiness" that I describe above.

---

> ### Author Response · Authors · 2025-06-03
>
> Thanks so much for your thoughtful review and constructive feedback. We'd like to address each concern below:
>
> Experiment 1 Clarification:
>
> Thank you for asking for clarification about our adaptation of the first experiment. Here's a picture representation of the first experiment's original setting: https://pubmed.ncbi.nlm.nih.gov/30834701/#&gid=article-figures&pid=fig-2-uid-1, that all pictures used the same carrier phrase format (Gentif __. translated to There is/are __).
> When we adapted this paradigm for LLMs, we changed the carrier phrases altogether rather than directly translating them. We were concerned that the repeating "there is/are" hundreds of times with different nouns might be too monotonic as an input to the LLMs. Therefore we adapted the stimuli to small paragraphs containing both plural and singular forms as we presented in the paper, hoping to provide more naturalistic linguistic context and to increase task variety. We acknowledge that the change could substantially affect the results and does raise reproducibility concerns, meaning that our comparison with the human study may not be exact. We will work on finding better adaptations for studies that involve visual stimuli while maintaining the fidelity to the original paradigms.
>
> Experiment 2 Error types:
>
> Thanks for asking for clarification for experiment 2. Following the original study design, we presented 4 types of errors in an interleaved manner. Each testing trial consisted of 12 correct sentences and 12 incorrect ones, with 3 sentences for each type (3x4 = 12). The detailed counts on each error type are provided in Table 8 in Appendix (p16). We provided only a brief description of the error types in Section 4.2.2 but failed to refer to Table 8. We will add a sentence to linking to this table and provide clearer descriptions of specific error types. Thanks for catching this oversight!
>
> Conclusion and Future Directions:
>
> Thanks for raising an excellent point about the confusing statement in our conclusion. The sentence about using "LLM models as valuable computational implementations of certain cognitive theories" is indeed inconsistent with our findings, especially as the opening sentence in section 6.2.
> One of our goal was to investigate whether human implicit learning research could inform LLM ICL research, not the reverse. Given that the models clearly behaved differently from humans in several tasks, suggesting their use as cognitive models would be premature. We will remove this sentence and replace it with one that better aligns with our actual findings and research direction.
>
> Copyright Issues:
>
> We truly appreciate you bringing this important issue to our attention. We were not aware that the figures were copyrighted and we should definitely not use them without the authors' consent. We will remove them and create new visualizations base on the original data. Thanks for flagging this ethics concern!
>
> Literature additions:
>
> Is Furlong et al 2022 paper "Learning a Language from Inconsistent Input: Regularization in Child and Adult Learners" (Austin, Schuler, Furlong & Newport 2022)? Yes, it's a great summary on implicit learning and we will include it in Section 2.1.
> Is the rule generalization framework in Schuler 2017 the Tolerance Principle? We will mention it in Section 2.1 too!
>
> In-context learning explanation:
>
> You're absolutely right that we need to better explain in-context learning for a cognitive science audience. We will add a dedicated paragraph in Section 2.2 explaining what ICL means in the LLM context.
>
> Thanks again for your thorough review! We really appreciate it!

---

### Official Review · Reviewer_FBZf · 2025-05-13

**Rating:** 6
**Confidence:** 4
**Ethics Flag:** 1

**Summary:**

This paper presents an exploratory study of how LLMs might generalize implicitly from exposure to patterns when performing in-context learning as compared to humans’ implicit learning of the same patterns. The study consists of replicating three psychological studies (1. Schuler et al. (2016), 2. Valian & Coulson (1988), 3. Alamia et al. (2020)) using LLMs as the subject instead of humans as used in the original studies. Two LLMs are tested: GPT-4o (temperature 0) and GPT-o3-mini.

The general conclusion of this study is that the alignment of human and in-context LLM implicit learning varies across the studied tasks. The main contribution of the paper is the recreation of these studies using LLM subjects, and the comparison against the original human-subject results.

These three psychological studies function by presenting the subject with a set of training examples, followed by a set of testing examples, and finally followed by a “post-testing” prompt that elicits reasoning patterns used by the model. 1. and 2. focus on morphosyntactic agreement, and 3. focuses on learning a finite-state grammar (FSG) and uniquely involves dynamic feedback during the testing phase.

The psychological studies are recreated as originally performed on humans except for the exact prompt used for testing models and 1. which relies on a verbalization of the context as opposed to an image used in the original study.

In a little more detail, 1. studies agreement in grammatical number for nonce words; 2. studies a simple rule-based grammar; and 3. studies an FSG grammar.

**Questions To Authors:**

Suggestions:
- I am not sure it makes sense to binarize the concept of “similar to humans” in Table 1.
- It might make sense to compare more explicitly with [2], which is briefly mentioned in the related work, given the claim on line 29 “the process by which pre-trained models rapidly acquire structured linguistic knowledge during inference - when presented with limited examples - remains notably understudied”
- The distinction between the first study as morphological and the second as morpho-syntactic is not clear to me given that the first also depends on syntactic agreement.
- Line 45: “While comparing reasoning-oriented and general-purpose models is not a primary goal of this study” It might make sense to remove this line given that the paper continually compares a reasoning and general-purpose model. It therefore does seem like a goal of the paper.

Small typo: Line: 328 dropped a )

[2] Why Can GPT Learn In-Context? Language Models Secretly Perform Gradient Descent as Meta-Optimizers (Dai et al., Findings 2023)

**Reasons To Accept:**

- The recreation of the three studies appears valid. The methodology is sound insofar as we accept the original studies’ formulations.
- The description of methods is mostly clear.
- The final results were not predictable given the experiment setup, and contribute to a growing body of work studying the mechanisms of in-context learning.
- The results present open questions that would be interesting to investigate further in future work such as discrepancies between the 4o and o3 models.

**Reasons To Reject:**

- The scope of the contribution is limited. The paper presents no theoretical hypotheses, and the three learning problems are taken from existing work. The paper itself summarizes this limitation well on line 368: “This study remains exploratory, deliberately avoiding commitment to specific theoretical hypotheses at this stage. Future targeted studies should also build on our exploratory findings, formulating and testing explicit hypotheses concerning specific linguistic phenomena.”
- The post-testing experiments could benefit from explicit justification given that the validity of LLM self-explanations is contested [1].
- The statistical power of the results is not clear to me. Statistical significance is presented for the main results being compared between humans and LLMs in the appendix, but the details of this testing is not clear. The number of LLM model “runs” is set to match the number of human participants in the original study, although the reason why is not motivated, and the limited number of LLM inferences will limit the statistical power.

[1] Are self-explanations from Large Language Models faithful? (Madsen et al., Findings 2024)

---

> ### Author Response · Authors · 2025-06-03
>
> Thanks so much for your thoughtful review and constructive feedback. We'd like to address each concern below:
>
> 1. Limited Scope and Lack of Theoretical Hypotheses
>
> You raise an important point about the exploratory nature of our work. We chose this approach because we hoped to use this exploratory investigation to identify what meaningful hypotheses might be formed, and to explore how human implicit learning studies could guide our understanding of LLM's ICL mechanisms.
> To our knowledge, this is among the first studies to replicate human implicit learning experiments to investigate LLM's ICL. Given that there is limited consensus about the mechanisms in either human implicit learning or LLM's ICL, we felt it was important to first establish where these systems behave similarly and where they differ, which could then inform more targeted hypothesis formation.
> We believe our study did reveal some promising patterns. For example, in Experiment 1, we found that the o3 mini model showed similar frequency effects in statistical learning as human adults. This finding could lead to a hypotheses about whether reasoning models apply statistical learning in ICL for morphological tasks.
>
> We hope this exploratory approach allows future work to build on these initial patterns to develop more targeted hypotheses. We see our contribution as providing a methodological foundation and initial empirical observations that can guide more hypothesis-driven research.
>
> 2. Post-testing Experiment Validity
>
> We appreciate this important methodological concern about LLM self-explanation validity. Thanks for bringing up the paper[1] about faithfulness of LLM. We were aware of the potential issues with LLM confabulation and tried to address this by not relying on post-testing explanations as sole evidence for rule learning. Instead, we compared models' self-explanations with the their actual performance on the tasks to look for alignment. For example, in experiment 1 (line 171-178) we calculated the correlation between pattern recognization in LLMs' self-explanation with their behavior in the testing set and found a close alignment.
>
> We positioned the post-testing self-explanation results as one additional lens into model behavior rather than definitive evidence of internal processes. Our primary conclusions about learning were based on performance data and statistical patterns, with the explanations serving as supplementary evidence when they aligned with behavioral observations. We agree this concern deserves more prominent discussion in the paper and will add clearer caveats about the limitations of self-explanations.
>
> 3. Statistical Power and Testing Details
>
> For the statistical testing, we closely replicated the statistical testing procedures in the three original studies. We recognize that more detail in our main text would be helpful. We will expand the statistical descriptions beyond the appendix.
> Regarding sample sizes, we matched the LLM runs with the number of human participants in the original studies following standard practice in comparative cognitive studies. This approach allows for direct statistical comparison using the same analytical framework. We acknowledge that limited samples could affect statistical power, but following the same statistical testing in the original studies with limited human samples, we were able to observe substantial effect sizes that allowed us to detect significant differences.
>
> Suggestions:
>
> 1. Thank you for this suggestion. We can see how the "similar to humans" categorization might oversimplify the nuanced patterns we observed. We will revise the table.
>
> 2. Thanks for highlighting this connection. While we considered discussing this work more extensively given the experimental similarities, we focused on artificial language tasks to isolate linguistics structures from semantic and pragmatic factors. The cited work (2) examined real language context, which introduce additional variables that could complicate interpretations of the learning mechanisms. Additionally, since our goal was to compare human and LLM performance on identical tasks, we needed studies that included human baselines, which the cited work did not provide.
>
> 3. Thank you for asking clarification on this distinction. The first study follows the traditional "wug test" paradigm used in morphological research. While it involves plural forms which has the component of syntactic agreement, all the test stimuli are presented with plural agreement, so participants (humans or LLMs) only need to learn the morphological inflection patterns rather than the agreement rules themselves. This focuses the learning challenge on morphological creativity rather than syntactic systematically, which is why we categorized it as primarily morphological.
>
> 4. We agree that it would make sense for use to delete this line since we did make systematic comparisons between reasoning and general-purpose models.
>
> Thanks again for your thorough review!

---

> > ### Comment · Reviewer_FBZf · 2025-06-05
> >
> > Thank you for this detailed clarification. I have updated my score particularly in response to the additional details that the authors' have noted will be added to the paper.

---

### Decision · Program_Chairs · 2025-07-08

**Decision:**

Accept

**Comment:**

The paper received three positive reviews. Synthesizing the reviews, the ensuing discussion, and my reading, I believe the paper has the following strengths and weaknesses relevant to the overall decision.

Reasons To Accept:
* The experiments and methods are clearly presented [FBZf,CGzC]
* Methodology appears valid [FBZf]
* Study contributes to study of the mechanisms of in-context learning.[FBZf,CGzC]
* The topic is interesting [CGzC]


Reasons To Reject:
A good number of the reasons to reject provided in the initial reviews have been addressed by the authors, or do not pose major concerns. I did consider the following three issues in my overall assessment:
* The scope is limited to an exploratory study using three learning problems from existing work and two models (both OpenAI), without presenting or investigating a theoretical hypothesis [FBZf]. I concur that this is a weakness.
* Reproducibility concerns due to the choice of how the image representation is replaced with text were raised by [FRjt] In the relevant study (Experiment 1), the paper uses 13 varied paragraphs in each condition. It seems that the paragraphs are matched between conditions, ruling out the potential that random differences between the two sets of 13 paragraphs drive (or mask) apparent differences between conditions. Overall, after consulting with the reviewers, I do not believe this to be a major concern.
* Only two models are tested (both OpenAI), limiting generalizability of take-aways from the results. [CGzC] To be fair, this may reflect resource constraints on the side of the authors.

**This paper went through ethics reviewing. Please review the ethics decision and details below.**
Decision: All good, nothing to do  or only minor recommendations